# 3D-Printed Hydrogels from Natural Polymers for Biomedical Applications: Conventional Fabrication Methods, Current Developments, Advantages, and Challenges

**DOI:** 10.3390/gels11030192

**Published:** 2025-03-09

**Authors:** Berk Uysal, Ujith S. K. Madduma-Bandarage, Hasani G. Jayasinghe, Sundar Madihally

**Affiliations:** 1School of Chemical Engineering, Oklahoma State University, 420 Engineering North, Stillwater, OK 74078, USA; berk.uysal@okstate.edu; 2Department of Chemistry, New Mexico Institute of Mining and Technology, Lopez Hall 221, Socorro, NM 87801, USA; ujith.maddumabandarage@nmt.edu; 3Mathematics, Physical and Natural Sciences Division, University of New Mexico-Gallup, 705 Gurley Ave., Gallup, NM 87301, USA; hjayasinghe@unm.edu

**Keywords:** 3D printing, natural polymers, hydrogels, biomedical applications

## Abstract

Hydrogels are network polymers with high water-bearing capacity resembling the extracellular matrix. Recently, many studies have focused on synthesizing hydrogels from natural sources as they are biocompatible, biodegradable, and readily available. However, the structural complexities of biological tissues and organs limit the use of hydrogels fabricated with conventional methods. Since 3D printing can overcome this barrier, more interest has been drawn toward the 3D printing of hydrogels. This review discusses the structure of hydrogels and their potential biomedical applications with more emphasis on natural hydrogels. There is a discussion on various formulations of alginates, chitosan, gelatin, and hyaluronic acid. Furthermore, we discussed the 3D printing techniques available for hydrogels and their advantages and limitations.

## 1. Introduction

Hydrogels have gained significant attraction in the biomedical field due to their unique physical and chemical properties and versatility. Hydrogels are extensively used with stem cells to study cell adhesion, morphology, migration, proliferation, and differentiation on materials [1,2,3,4,5,6]. The physiochemical tunability of hydrogels supports in-depth investigations of the effect of cellular-matrix interactions on cell responses and determining cell fate [7,8]. Moreover, hydrogels have become excellent candidates for exploring potential applications in drug or cell delivery systems, scaffolds in tissue engineering, and wound dressings [9,10,11,12,13,14]. However, limitations in traditional fabrication techniques and the complexity of biological systems restrict the use of hydrogels in biomedical applications. Additive manufacturing (also known as 3D printing) can overcome the weaknesses of traditional methods. Even though 3D printing has an excellent potential to fabricate 3D hydrogel architectures, more studies must be conducted to harness the full potential of the 3D printing technology. This review covers the fundamentals of hydrogels and 3D printing technology, emphasizing the limitations of traditional fabrication methods, the advantages of replacing 3D printing with conventional methods in producing hydrogels, recent advancements in 3D printing technology, and the challenges of 3D printing.

## 2. Hydrogels

Hydrogels are three-dimensional polymer networks that absorb large amounts of water without dissolving. The crosslinks between the polymer chains create the network structure. Figure 1 shows the structure of a hydrogel. These networks can absorb water up to hundreds or thousands of times their dry weight while retaining structural integrity [15]. The hydrophilic groups in the polymer network are responsible for the water absorptivity, and the crosslinking between polymer chains resists the dissolution [16]. Other than water absorptivity, hydrogels have many interesting chemical and physical properties, such as tunable mechanical properties, degradability, chemical stability, porosity, and stimuli responsiveness. Hydrogels can be synthesized with various functional groups to introduce and tune the mechanical, chemical, and biological properties. The number of crosslinks in a unit volume of a hydrogel, also known as the crosslinking density, determines the mechanical properties of the resultant hydrogel. Therefore, hydrogels with mechanical properties desirable for a specific application can be easily prepared without significantly affecting the chemical composition or the fabrication conditions [17]. Incorporating functional groups susceptible to hydrolysis or biological degradation enhances the degradability of hydrogels. Varying the concentration of these functionalities enables the fabrication of hydrogels with tunable degradability [18]. Furthermore, incorporating chemical groups that can respond to external stimuli such as temperature, pH, or chemicals introduces stimuli responsiveness to the hydrogel [19,20,21,22,23]. The tunability of these properties in a vast range allows researchers to design materials for many applications, such as biomedical, cosmetic, environmental, agricultural, and catalytic applications [24]. The following subsections describe the classification, fabrication methods, biomedical applications, and limitations of hydrogels in detail.

### 2.1. Classification

Hydrogels are classified into several groups based on their characteristic features. Figure 2 shows a classification scheme for hydrogels. The spheres in Figure 2 represent the monomers, and the bond between monomer units is depicted in red. The spheres represent the monomer units, whereas the identical color spheres represent the same monomer unit, and different color spheres represent different monomer units. Hydrogels have a three-dimensional crosslinked polymer network. Hydrogels can be prepared by crosslinking using natural or synthetic polymers. The crosslinking between polymer chains can be either physical or chemical. Polymers physically crosslink each other with hydrophobic interactions, hydrogen bonds, ionic interactions, or crystallinity [25]. In chemical crosslinking, the functional groups present in the polymer chains form covalent bonds. Chemically crosslinked hydrogels have more stability, extended durability, and better mechanical properties than physically crosslinked hydrogels [26].

Hydrogels can also be categorized into two main groups: natural or synthetic, depending on the origin of the source. Natural hydrogels are prepared using compounds extracted from natural sources such as plants, animals, and microorganisms, whereas synthetic hydrogels are made from synthetic monomers. Natural polymers, including chitin, cellulose, gelatin, alginate, and agarose, can be crosslinked via chemical or physical crosslinking to produce hydrogels [27]. Most other hydrogels are synthetic in origin. Another feature used in the hydrogel classification is polymer characteristics. Polymers may have crystallinity depending on their stereochemistry (isotactic, syndiotactic, and atactic). Therefore, most hydrogels are amorphous or semi-crystalline. Physically crosslinked polyethyleneimine (PEI) is an example of a crystalline hydrogel [28]. Meanwhile, physically crosslinked PDMAM-co-PMEA-g-PCL gives a semi-crystalline hydrogel [29]. Based on the charge, hydrogels have three classes: ionic, nonionic, and zwitterionic. Ionic polymers are further categorized as cationic or anionic. Ionic polymers are used to synthesize ionic hydrogels. Some examples of ionic polymers are poly(acrylic acid) (PAA), poly(methacrylic acid) (PMAA), poly(diethylaminoethyl methacrylate) (PDEAEMA), and poly(dimethylaminoethyl methacrylate) (PDMAEMA) [30]. These are represented in Figure 2 by the spheres with the charge. Zwitterionic hydrogels are prepared by using zwitterionic monomers such as 3-[[2-(methacryloyloxy)ethyl]-dimethylammonio]propane-1-sulfonate, 3-[[2-(methacryloyloxy)ethyl]dimethylammonio]propionate, 2-Methacryloyloxyethyl, and polysulfobetaine (pSBMA), 3-[(3-acrylamidopropyl)dimethylammonio]propanoate [31,32].

The polymer chain composition is another factor used to classify hydrogels. Polymers used in hydrogel fabrication are homopolymers, heteropolymers, copolymers, interpenetrating polymer networks (IPNs), semi-interpenetrating polymer networks (semi-IPNs), or polymer composites. Homopolymers consist of one type of monomer, and copolymers contain two or more types of monomers. Some examples of homopolymers are Poly(hydroxyethyl methacrylate), Poly(glyceryl methacrylate), and Poly(hydroxypropyl methacrylate). These polymers can be crosslinked by using Triethylene glycol dimethacrylate or 1,1,1-trimethylolpropane trimethacrylate [31]. Depending on how these monomers are linked, copolymers are classified as block, alternative, random, or graft polymers. In IPNs, two or more polymer networks are partially interlacing on the molecular scale [33], whereas semi-IPNs contain an embedded linear polymer chain in their polymer network [34]. An example of an IPN hydrogel is a hydrogel made using poly(N–isopropyl acrylamide-co-acrylic acid) and Poly(ethylene glycol) monomers and MBAm as the crosslinker [35]. A representation of an IPN hydrogel is shown in Figure 2, where the two color chains, blue and green, represent two polymer chains. Composite hydrogels have nanomaterials (represented as a red sphere in Figure 2) incorporated into their polymer network and have unique chemical, physical, biological, and electrical properties [36].

Stimuli responsiveness and physical aspect are two other features used to classify hydrogels. Hydrogels may respond to external stimuli, such as biological, physical, or chemical stimuli, by changing their volume—swelling or deswelling. Stimuli-responsive hydrogels are mostly used in sensing applications. The physical appearance of hydrogels can be sol–gel, matrix, micro/nanoparticles, and films. Hydrogels can undergo simple phase transitions and transform into a solution. Hydrogels become a flowing fluid in the solution phase. Since these hydrogels can be injected into a defective site, sol–gel hydrogels are suitable for biomedical applications. When fabricated as particles, they have a higher potential for use in drug delivery systems. Hydrogel films developed as membranes or coatings have applications in devices associated with biological systems.

### 2.2. Conventional Fabrication Methods of Hydrogels

#### 2.2.1. Crosslinking in Hydrogels

In a solution, crosslinking between polymer chains results in soluble branched polymers (sol phase). These branched polymers grow as the crosslinking increases and eventually become insoluble, producing a three-dimensional network (gel phase). This process is called the sol–gel transition or gelation, and the critical point where the sol phase transitions to the gel phase is called the gel point. Gelation is a result of either physical or chemical crosslinking. The physical gelation process produces either weak or strong gels. Strong gels have glassy nodules, double/triple helices, or lamellas. Meanwhile, weak gels have hydrogen bonds, ionic or hydrophobic interactions, and aggregation. Chemical gelation occurs due to condensation, addition, or chemical crosslinking [37].

The methods to produce physically crosslinked hydrogels are heating–cooling cycles, ionic interactions, complex coacervation, hydrogen bonding, heat-induced aggregation, and freeze-thawing. Cooling polymer chains in a hot polymer solution may form helices, which further aggregate in the presence of salts [37,38]. Two main examples are gelatin and carrageenan gels [39]. Ionic polymers form crosslinks in the presence of di- or trivalent counterions; for example, alginate-based polymer chains crosslink when Ca^2+^ ions are present [37]. A mixture of an anionic polymer and a cationic polymer results in a complex coacervate gel, e.g., coacervate of polyanionic xanthan with polycationic chitosan [40]. Hydrogen bonding is another method for producing crosslinked polymer networks or gels. Polymers carrying carboxylic groups, such as carboxymethyl cellulose, can produce hydrogen bonds when the pH of the medium is low [41]. Heat treatments can induce the aggregation of protein-containing components of polymers having proteinaceous groups to produce a hydrogel [42]. Freeze-thawing of polymers may form microcrystals, resulting in a hydrogel. For example, xanthan gum and polyvinyl alcohol hydrogels can be formed by the freeze-thawing method [43].

Chemical crosslinking involves reactions between readily available functional groups in polymer chains, grafted monomers of the polymer backbone, or polymer chains with crosslinking agents. Functional groups such as amines, carboxylic acids, or hydroxyls can react to form covalent linkages between polymer chains. Aldehydes can crosslink polymers having hydroxyl groups, e.g., polyvinyl alcohol crosslinked through glutaraldehyde [44]. Another way to synthesize hydrogels is through addition reactions between crosslinking molecules and functional groups present in the polymer backbone [25]. High energy radiation, such as gamma rays and electron beams, may lead to crosslinking between unsaturated polymers [45]. Also, free-radical polymerization of polymerizable groups in the polymer backbone can produce hydrogels.

#### 2.2.2. Fabrication Methods of Hydrogels

Different techniques have been used to fabricate hydrogels based on their application. For example, the methods used in the fabrication of hydrogel scaffolds are emulsification, lyophilization (freeze-drying), solvent casting–leaching, gas foaming–leaching, electrospinning, photolithography, microfluidics, and micro molding [6].

*Emulsification* is a standard method that produces hydrogel nano- and microparticles. Agitation of a multi-phase mixture results in aqueous droplets containing hydrogel precursors. Lyophilization is a technique where rapid cooling separates the phases, followed by solvent sublimation. Lyophilization produces porous hydrogel matrices, and emulsification with lyophilization results in interconnectivities between pores. Other techniques used to fabricate porous hydrogels are solvent casting–leaching and gas foaming–leaching. Both of these methods use salts to generate pores. The solvent casting–leaching method uses particulate salts with specific dimensions. With the evaporation of the solvent, these salt particulates trap inside the hydrogel, and the dissolution of the salt particulates in an aqueous media results in pores. This method produces hydrogels with uniform pore sizes. In the gas foaming–leaching technique, the salt used produces a gas, e.g., ammonium bicarbonate, which produces ammonia and carbon dioxide. The leaching of these gases creates a porous matrix. Electrospinning is another method that makes interconnected porous scaffolds. An external electric field is applied to generate microfibers through a capillary tube. Here, a high voltage charges the polymer, and then a thin jet filament is drawn toward an oppositely charged plate or rotating collector [46].

*Photolithography* is another method used for fabricating hydrogels. Exposure of a thin film of a photocrosslinkable polymer to UV light through a mask results in a hydrogel with a pattern. Since the UV light only passes through the transparent areas of the mask, the hydrogel has a negative pattern [47]. Photolithography fabricates micro-engineered hydrogel scaffolds. However, this method has significant drawbacks, including the use of UV light and harmful photoinitiators. Soft lithography is another method that can produce structured hydrogels. Here, a negative pattern printed on an elastomer such as poly dimethyl siloxane (PDMS) is used as a mold to cast the hydrogel by transferring a microstructure fabricated on a silicon wafer [48,49,50].

The techniques such as cryogelation, freeze-drying, gas foaming–leaching, microemulsion formation, and porogen leaching result in porous gels. These processes offer limited control on pore size distribution which is useful in tissue regeneration applications. However, drug delivery applications need hydrogels with specific designs, as these gels are administered to patients through various routes. For example, to inject a macroscopic gel into a patient, the gel should undergo either sol–gel transformation, shear-thinning, or collapse when the water is squeezed out of it [51]. The drug delivery system is injected as a liquid for gels that require in situ gelling. The liquid form undergoes a sol–gel transformation and transforms into a gel inside the injection site. The sol–gel transition may occur due to a charge interaction [52], Michael addition [53], and stereocomplexation [54]. Shear-thinning is another method used to inject pre-gelled hydrogels into the body. Under shear stress, the hydrogel flows as a low-viscous fluid, and after removing the shear stress, the hydrogel returns to its former status.

Deposition of layers of cell-loaded microgels produces hydrogels with complex structures. The microfluidic technique can make cell-loaded microgels. Here, a mixture of hydrogel precursors and cells is sent through microfluidic channels. These channels control the shape of the resulting hydrogel [55]. Micromolding is another method that produces microgels. Emulsion and nanomolding are used for the fabrication of nanogels [51]. Micromolding and nanomolding can produce hydrogels with controlled size and porosity. In these techniques, the hydrogel precursor solution is poured into molds of the desired size, followed by crosslinking. Methods such as soft lithography allow for the easy fabrication of molds. In this technique, stamps with the desired sizes are used [56,57].

### 2.3. Hydrogels in Biomedical Applications

Biomaterials require specific features such as higher biocompatibility, tunable biodegradability, and desirable mechanical properties. Hydrogels have gained more attraction as biomaterials as they can be designed to occupy the desirable properties of biomaterials. Hydrogels have many potential applications in the biomedical field, such as tissue engineering, drug delivery, and wound healing.

Since hydrogels resemble the extracellular matrix, they are excellent candidates in tissue engineering. Moreover, hydrogels have a framework that facilitates cellular proliferation and survival. Tissue engineering aims to restore, preserve, or improve tissue functions, where biologically active scaffolds play a significant role. The basic requirements for scaffolds are biodegradability, biocompatibility, desired pore size, shape, and volume, pore interconnectivity, tissue-specific mechanical characteristics, relatively large and accessible surface area (for cell attachment), and facilitation for vascularization. The major challenges in tissue engineering are vascularization, tissue architecture, and simultaneous seeding of multiple cells. Therefore, controlling the scaffold’s porosity, shape, size, and surface morphology is crucial [6]. Recent advances in hydrogel fabrication open pathways to developing scaffolds with desired features. Hydrogel scaffolds provide the desired bulk and mechanical structures to the newly developing tissue. In addition to scaffolds, hydrogels can be used as carriers for cell transplantation as they can immunoisolate themselves without interfering with the diffusion of nutrients, oxygen, and metabolic products [6]. Also, the hydrogel can act as a barrier to avoid restenosis or thrombosis due to postoperative adhesion formation [58].

Conventional methods used in drug administration have several drawbacks. For example, a higher dosage or repeated administration is needed to have a therapeutic effect. However, these drawbacks lower efficiency and cause severe side effects and toxicity. Controlled drug delivery can overcome the issues of conventional methods. A drug delivery system can control the release of a drug to a targeted site. Usually, these systems contain nanoparticles, membranes, liposomes, and hydrogels [51]. Hydrogels are excellent candidates for drug delivery applications due to their high biocompatibility, easy encapsulation of drugs, stability, matching mechanical properties with tissues, protection of the drug from body enzymes, and the possibility of designing with different physical aspects. For example, the sol–gel transformation of hydrogels allows for administration via an injection.

The skin has the potential to regenerate. However, wounds larger than a certain diameter require skin transplants. Treatments include skin grafts and skin flaps, skin expansion, and dermal substitution. However, these techniques have several disadvantages, i.e., donor site shortage and hypertrophic scars or keloids [11]. Therefore, tissue-engineered skin substitution is a suitable replacement for the above methods. Among the materials studied for tissue-engineered skin substitutes, hydrogels have gained more attention due to their ability to mimic the native skin microenvironment. Since hydrogels allow oxygen diffusion, maintain high moisture content, and absorb wound exudates, they promote rapid healing [59]. Novel techniques will enable the production of sprayable wound dressings [60,61]. These are in situ forming gels with simple application and a low price.

### 2.4. Limitations of Hydrogels

Not all hydrogels have the desired properties needed for a biomedical application. Some hydrogels have poor biocompatibility, biodegradability, or both. Modifications on constituent polymers may increase the biocompatibility and biodegradability of a hydrogel. For example, poly(N-isopropylacrylamide) (PNIPAm) can gelate reversibly at higher temperatures and be used to prepare in situ-gelling hydrogels. However, PNIPAm has poor biodegradability. Incorporation of poly(ethylene glycol) (PEG) or poly(ε-caprolactone) (PCL) can increase the biodegradability of PNIPAm [62]. Some other challenges associated with hydrogels are the lack of desired mechanical properties, poor physical stability—especially in a physiological environment—slow responsiveness, difficulties in binding and releasing drugs—in drug delivery systems—and harsh conditions required in the fabrication process. Solutions for these problems involve modifications in constituent polymer chains, introducing additional crosslinks, making blends with nanoparticles, making thinner hydrogels, and incorporating copolymerization with different polymers [51,63].

## 3. 3D Printing

Three-dimensional (3D) printing, also known as additive manufacturing, uses layer-by-layer fabrication technology to create 3D objects. In this technique, a material is added as successive cross-sectional layers [64]. The common materials used for 3D printing are ceramics, polymers, metals, composites, and glass [65,66]. The layer addition is computer-guided, and the final object is a replica of a 3D computer-generated model or an actual object scanned as a computer-aided design file (CAD) [64]. Figure 3 shows the general steps of 3D printing. Additive manufacturing offers advantages over traditional methods, including design flexibility, the low cost associated with geometric complexity, dimensional accuracy, single-part assembly, and cost and time efficiency [64]. 3D printing has applications in many industries, such as in the automotive, aerospace, food, construction, fabric, fashion, electric, electronic, healthcare, and medical industries [66]. The use of 3D printing in medical applications is growing fast. The human body has structures with complex shapes unique to each person. Since 3D printing can produce complex geometric shapes, this technology is valuable for biomedical applications. As defined by ISO/ASTM 52900-2015, there are seven types of additive manufacturing: binder jetting, directed energy deposition, materials extrusion, materials jetting, powder bed fusion, sheet lamination, and vat photopolymerization [66,67].

The material extrusion technique uses heated material, usually a plastic filament, extruded through a nozzle. Fused deposition modeling (FDM) is an example of a material extrusion method. Material extrusion printing uses molten or semi-molten polymers, polymer solutions, dispersions, or pastes, and the material is deposited as layers on top of each other using a movable nozzle. The ink is dispensed pneumatically (by altering gas pressure), pushed with a piston, or screwed [68,69,70]. Direct energy deposition is similar to the material extrusion technique. However, in the material extrusion method, the nozzle is fixed, while in direct energy deposition, the nozzle can move in multiple directions. A laser or an electron beam melts the printing material. Typically, metals are used as the material (as a wire or in powdered form) in indirect energy deposition. Laser deposition and laser-engineered net shaping are examples of this technology [71].

Binder jetting uses powder particles joined by selectively depositing a liquid binding agent. In materials jetting, build and support materials are selectively sprayed as droplets and cured with ultraviolet light or heat. Material jetting is a similar technique as 2D inkjet printing. Jetting can be either continuous or drop on demand. In powder bed fusion, a powdered material is melted and fused using either a laser or an electron beam. The processes involved in sheet lamination are ultrasonic additive manufacturing and laminated object manufacturing. Sheets or ribbons of metals are bound together using ultrasonic additive manufacturing. The laminated object manufacturing uses paper, which is bound together using an adhesive [71,72]. Vat photopolymerization uses a vat of a liquid polymer resin that is cured, using ultraviolet light, layer by layer only at the required places to construct the 3D object. In this technology, the platform will move down as each layer is built. Vat photopolymerization produces 3D prints with higher resolution, greater efficiency, good surface finish, and printing accuracy. Examples of the vat polymerization technique are stereolithography, mask-projection vat photopolymerization, and two-photon polymerization [73].

### 3.1. 3D Printing of Hydrogels

The uniqueness of 3D printing is that it allows for the fabrication of objects with high precision, customization, predefined organization, reproducibility, and a range of materials [74]. Hydrogels have great potential in biomedical applications because of their customizable biological, physical, and chemical properties. Therefore, using hydrogels as a material for 3D printing makes it possible to fabricate patient-specific constructs with customized properties. Since hydrogels are sensitive to harsh conditions, not all printing techniques are suitable. Some standard methods used for the 3D printing of hydrogels are laser-based systems, material extrusion techniques, and material and binder jetting (Figure 4) [74].

*Laser-based systems* use light energy in predefined patterns to crosslink photocrosslinkable polymers to produce a hydrogel with a desired shape. Two-photon polymerization, stereolithography, and laser-induced forward transfer are some techniques that fall under laser-based approaches [74]. In stereolithography, a laser source, i.e., UV light, induces polymerization and crosslinking on desired spots of a liquid material while the build platform moves along the vertical plane. Once the fabrication is completed, the residual liquid is washed off, and the printed object is cured with UV light to complete the process [75,76]. In two-photon polymerization, a photoinitiator absorbs two photons of 800 nm wavelength produced by a near-infrared femtosecond laser and acts as one 400 nm photon and starts the polymerization [77]. Laser-induced forward transfer uses a pulsed laser beam focused onto a hydrogel through a donor substrate, i.e., laser-transparent quartz. The laser-absorbed hydrogel is propelled and deposited on a receiving substrate as a voxel. Here, the 3D object is printed as a combination of voxels [78]. Material extrusion techniques and material and binder jetting are discussed in detail under Section 3 (3D printing).

3D bioprinting is a recently adapted additive manufacturing in which cell-laden biomaterials, also known as bioinks, are deposited layer by layer to fabricate living tissues or organs. Techniques such as X-ray imaging, magnetic resonance imaging (MRI), and micro-computerized tomography scans (μ-CT-scans) can produce complex geometric data helpful in designing CAD files of tissues. 3D printing can create complex constructions with high accuracy. Therefore, when the printing material is integrated with living cells, living tissue can be printed [79]. Hydrogels, as the printing materials, are ideal candidates for producing cell-laden biomaterials. Some examples of natural hydrogels used to design cell-laden biomaterials are agarose, alginate, gelatin, chitosan, collagen, hyaluronic acid, fibrin, keratin, and their composites [80,81,82,83,84]. Dynamic hydrogels have self-healing properties, self-recovering properties, or both. Self-healing hydrogels can rejoin themselves to regain their original shape after damage. Self-recovery is another vital feature. The capability of a rheologically deformed hydrogel to recover internal damages and regain its original form is called self-recovery [85]. Static hydrogels show poor self-recovery properties due to the processing of hydrogels/polymers during 3D printing. Since dynamic hydrogels can recover fast, they are much more suitable for 3D printing. Dynamic hydrogels have dynamic covalent or non-covalent interactions. The covalent interactions include the Diels–Alder reaction, acylhydrazone bonds, imine bonds, and disulfide bonds. The non-covalent interactions are hydrophobic, hydrogen bonds, ionic bonds, and host–guest interactions [86,87]. Some polymers used to synthesize self-healing hydrogels are polyacrylamide, polyvinyl alcohol, polyethylene glycol, gelatin, chitosan, and alginate. These hydrogels have a vast range of applications, i.e., soft robots [88,89], wound healing [90,91], tissue engineering [92,93], surface coating [94,95], and drug/cell delivery [96,97].

A few years back, including the fourth dimension—time—a new concept of 4D printing was introduced. The materials printed using 4D printing can change their shape with time [98]. Shape-memory alloys, shape-memory polymers, and hydrogels are the active materials used for 4D printing. Since hydrogels can change their volume in response to an external stimulus, they are suitable for 4D printing. In addition to modifying the material, the 3D printing techniques should also be modified to print 4D architectures. Adding air circulation systems for the FDM method is an example [99].

### 3.2. 3D-Printed Hydrogels from Natural Polymers

Natural hydrogels are gaining increasing attention for biomedical applications due to their proven biocompatibility, biodegradability, low or nontoxicity, abundance, and cost-effectiveness [100], despite the advantages of synthetic hydrogels, including high gel strength, high water absorptivity, and longer shelf life [83]. Since 3D printing is an emerging technique used in the fabrication of biomaterials, various studies focus on the 3D printing of hydrogels from natural polymers, also known as biopolymers. Biopolymers used in hydrogel synthesis mainly include polysaccharides and proteins [100]. Commonly used polysaccharides are alginate, chitosan, hyaluronic acid, and pectin. Collagen, gelatin, and silk fibroin are examples of frequently prepared protein-based hydrogels [100]. These hydrogels can be used as drug delivery systems, scaffolds for engineering tissues, wound dressings, and bioinks [81,82,84,101,102]. Table 1 summarizes the 3D-printed hydrogels of natural polymer blends studied for tissue engineering applications.

Since hydrogels produced from one type of monomer may lack properties such as dimensional stability, mechanical strength, rheological properties, and biodegradability, researchers investigate the possibility of blending natural polymers with other natural and/or synthetic polymers to synthesize hydrogels with desired properties [103,104,105]. For example, the mechanical properties of silk fibroin hydrogels were increased by adding hydroxypropyl methylcellulose (HPMC) [103]. Another study reports the preparation of silk fibroin gelatin blends to achieve high structural stability of hydrogels [104].

**Table 1 gels-11-00192-t001:** A summary of 3D-printed hydrogels of natural polymer blends studied for tissue engineering applications.

**Alginate-Based Hydrogel**				
**Alginate (Alg) Concentration**	**Blended with**	**Gelation Mechanism**	**Application**	**Reference**
2% *w*/*v*	Hyaluronic Acid 1% *w*/*v*	Ca^2+^ Ionic Crosslinking	Articular Cartilage	[106]
5% *w*/*v*	Chitosan 1–2 *w*/*w*—Hydroxyapatite 0.1–0.4 *w*/*w*	Ca^2+^ Ionic Crosslinking	Bone Tissue	[107]
6–10% *w*/*v*	Hydroxyapatite 0–24% *w*/*v*	Ca^2+^ Ionic Crosslinking	Bone Tissue	[108]
2% *w*/*v*	Nanocellulose: Alginate 8:2 *v*/*v*—Hyaluronic Acid 1% *w*/*v*	Ca^2+^ Ionic Crosslinking	Cartilage Tissue	[109]
3% *w*/*v*	Graphene Oxide 0.5 mg/mL	Ca^2+^ Ionic Crosslinking	Bone Tissue	[110]
0.1 g/mL	Collagen 15 mg/mL—Agarose 15 mg/mL: Alginate 1:4 *v*/*v*	Ca^2+^ Ionic Crosslinking	Cartilage Tissue	[111]
5% *w*/*v*	Poly(amino acid) 0–2% *w*/*v*	Ca^2+^ Ionic Crosslinking	Tissue Engineering Scaffold	[112]
**Chitosan-Based Hydrogel**				
**Chitosan (CS) Concentration**	**Blended with**	**Gelation Mechanism**	**Application**	**References**
2% *w*/*w*	Alginate 5% *w*/*w* and Gelatin 30% *w*/*w*—mixed 2:1:1 *v*/*v*/*v* Gel:Alg:CS	Ionic Crosslinking	Liver Tissue	[113]
3% *w*/*v*	Hyaluronic Acid 0–40% *v*/*v* with Chitosan	Ionic Interaction (NaOH and EtOH)	Bone Tissue	[114]
2.5 *w*/*v*	Gelatin 2.5–7.5% *w*/*v*	pH Crosslinking	Skin Tissue	[115]
2% *w*/*v*	Hyaluronic Acid 0–20 mg/mL	Thermal Gelation	Bone Tissue	[116]
2–4% *w*/*v*	Alginate 3–6% *w*/*v*	pH Crosslinking	Vascular Tissue	[117]
3.5–4.5% *w*/*w*	Dissolved in Alkali/Urea aqueous solution	Thermal Gelation	Wound Healing	[118]
2–4% *w*/*v*	Chitosan 2–4% *w*/*v*	Thermal Gelation	Tissue Engineering	[82]
**Gelatin-Based Hydrogel**				
**Gelatin Concentration**	**Blended with**	**Gelation Mechanism**	**Application**	**References**
5% *w*/*v*	Gelatin: Chitosan 10:1 ratio	3% sodium tripolyphosphate	Liver Tissue	[119]
20% *w*/*w*	Alginate 5% *w*/*w* mixed with gelatin at 3:7, 4:6, 5:5, 6:4, 7:3	Ca^2+^ Ionic Crosslinking	Vascular Tissue	[120]
Gelatin Methacrylate (GelMA) 5–20% *w*/*v*	-	Irgacure Photocrosslinking	Vascular Tissue	[121]
GelMA 5–7% *w*/*v*	Alginate 1–% *w*/*v*—4-arm poly(ethylene glycol)-tetra-acrylate 1–3% *w*/*v*	Photocrosslinking and Ca^2+^ Ionic Crosslinking	Vascular Tissue	[122]
10% *w*/*v*	Alginate 1%, 2%, 4% *w*/*v* Gelatin: Alginate 1:4	Ca^2+^ Ionic Crosslinking	Muscle Tissue	[123]
50% *w*/*w*	Alginate: Fibrinogen 25:25 *w*/*w*	Ca^2+^ Ionic Crosslinking	Vascular Tissue	[124]
20% *w*/*v*	Alginate 6% *w*/*v*—Fibrinogen 5% *w*/*v*—Gel:Alg:Fib 2:1:1 *v*/*v*/*v*	Ca^2+^ Ionic Crosslinking	Vascular Tissue	[125]
**Hyaluronic Acid-Based Hydrogel**				
**Hyaluronic Acid Concentration**	**Blended with**	**Gelation Mechanism**	**Application**	**References**
6 mg/mL Acetic Acid	Collagen 60 mg/mL Acetic Acid	Thermal Gelation	Tissue Engineering Scaffold	[126]
Methacrylated 1% *w*/*w*	GelMA 5% *w*/*w*	UV Crosslinking	Cartilage Tissue Repair	[127]
HA mono-aldehyde (30–70 mg/mL)	Carboxymethyl Cellulose—Carbohydrazide 30–70 mg/mL	Covalent Crosslinking	Vascular Tissue	[128]
0.5% *w*/*v*	Alginate 1% *w*/*v*—RGD Modified Alginate 1% *w*/*v*—Fibrinogen 20 mg/mL, 40 mg/mL	Covalent Crosslinking	Nerve Tissue	[129]
4 mg/mL	Fibrinogen 50 mg/mL—Factor XIII 1 U/mL—Aprotinin 0.5 mg/mL	Covalent Crosslinking	Nerve Tissue, Tissue Engineering Scaffolds	[130]
Methacrylated (2%, 4%, 6% *w*/*v*)	GelMA 6%, 10%, 12% *w*/*v*	Irgacure 2959 Photocrosslinking	Heart Valve Conduit	[131]
Methacrylated (1% *w*/*v*)	Arg-Gly-Asp-Ser (RGDS) peptide 2 mM/mL	UV Crosslinking	Retina Cell Culture	[132]

### 3.3. Requirements

The physicochemical properties of hydrogels dictate their suitability for 3D printing shapes using different techniques. The two fundamental factors determining printability are the rheological properties and gelation mechanisms. For example, hydrogels with high viscosity showed high printing accuracy [133].

The flow behavior of hydrogels, i.e., the rheological properties through nozzles, determine their suitability for 3D printing. These properties are mainly divided into viscosity, shear thinning, and yield stress [74]. Low viscosity during printing and sufficient mechanical strength afterwards is the most ideal condition to achieve. Rheological experiments are performed to assess the suitability of the proposed hydrogel for 3D printing. The rheological properties of hydrogels mainly rely on their microstructure which is dependent on the concentration of its components [74,134,135,136], different functional groups in the material, and crystalline properties. Thus, blends of various semicrystalline polymers such as alginates, chitosan, and gelatin are added to vary the ratio of the crystalline region to the amorphous region. This allows proposing hydrogels to be adjusted for 3D printing by changing the ratios if there is a deficiency. Synthetic semicrystalline polymers such as Poloxamer are also blended in some cases to adjust the rheological properties as necessary for 3D printing hydrogels using natural polymers.

Due to shear thinning properties, a decrease in viscosity occurs with an increasing shear rate [135]. Although a lower viscosity is preferred for 3D printing, too low of a viscosity would cause issues related to the flow of the fluid. A high G’ (storage modulus) value is necessary for shear thinning behavior to occur. When comparing G’ (storage modulus) and G” (loss modulus), G’ > G” is the criteria for gelation to happen. As a way to improve the printing ability of hydrogels, materials with faster recovery would be more suitable [137,138].

The relations between the printhead speed, volume flow rate of the material through the nozzle, and shear rate are influenced by microstructural properties that would change depending on the ratios and materials used, as well as knowing whether the crosslinks between polymers are reversible or not [138]. For Newtonian fluids under steady laminar flow conditions, the Poiseuille equation is derived using the conservation of momentum,$$p\left(\pi r^{2}\right)-\left(p+\Delta p\right)\pi r^{2}=2\tau \pi rL\;$$
where p is the pressure, τ is the shear stress, and L is the length [138,139]; reorganizing the expression, the shear stress profile can be obtained across the nozzle as$$\tau =\frac{-\Delta p}{2L}r$$

Calculating the shear stress values during the printing process helps determine the behavior of the hydrogels and determine their compatibility for the 3D printing process. However, most polymeric solutions are non-Newtonian and follow a power-law model, where shear stress is expressed as a function of the shear rate as$$\tau =K{\left(\dot{\gamma }\right)}^{n}$$
where *K* is called the flow consistency index, and *n* is called the flow behavior index. Both parameters are obtained via experimental curve-fitting data relating the shear stress or apparent viscosity with $\dot{\gamma }$, the shear rate. For Newtonian fluids, the value of *n* is 1, and for shear thinning fluids, the value of *n* is less than 1. Both the *K* and *n* values depict the microstructure characteristics of the solution and are dependent on the concentration, crystallinity, and molecular weight of the polymers. Both *K* and *n* are evaluated for various combinations and routinely reported for many combinations of polymers. However, there are variabilities in these values based on the shear rate used and the combination of other polymers. In any case, the apparent viscosity of a power-law fluid is written by comparing to Newtons law as$$µ=K{\left(\dot{\gamma }\right)}^{n-1}$$
which results in the final form of the shear rate:$$\dot{\gamma }=\frac{du}{dr}=\frac{-\Delta p}{2\eta L}r$$
where u is the flow velocity at r. The volumetric flow rate ($\dot{V}$) of a non-Newtonian fluid can be expressed as follows, with *R* as the radius of the container [138,139]:$$\dot{V}=\pi R^{2}V=\pi \left(\frac{n}{3n+1}\right){\left(\frac{-\Delta p}{2mL}\right)}^{\frac{1}{n}}R^{\left(\frac{3n+1}{n}\right)}\;$$
allowing the shear rate ($\dot{\gamma }$) to be described as follows:$$\dot{\gamma }=r\left[\frac{VR^{2}}{\left(\frac{n}{3n+1}\right)R^{\left(\frac{3n+1}{n}\right)}}\right]\;$$

Assuming that the volume of the hydrogel is constant before and after printing, the following ratios can be obtained [138,139]:$$\frac{{\dot{\gamma }}_{2}}{{\dot{\gamma }}_{1}}={\left(\frac{D_{1}}{D_{2}}\right)}^{\left(3+\frac{1}{n}\right)}\mathrm{and}\;V_{2}=V_{1}{\left(\frac{D_{1}}{D_{2}}\right)}^{2}$$
where ${\dot{\gamma }}_{2}$ is the shear rate in the syringe, ${\dot{\gamma }}_{1}$ is the shear rate in the nozzle, $V_{2}$ is the speed of the piston, $V_{1}$ is the speed of the extruded hydrogel, $D_{1}$ is the inner diameter of the syringe, and $D_{2}$ is the inner diameter of the nozzle [138,139].

Gelling properties also contribute to the printing quality by minimizing gravity effects. For instance, a hydrogel with a high gelation rate tends to produce a quality print. A high viscosity prevents the hydrogel from spreading after the deposition, and fast gelling solidifies the hydrogel quickly, avoiding any print deformations [140,141]. The gelation property is a function of the mechanism of gelation and the concentration of the polymer. Some utilize a bath of compatible density into which printing is performed [142,143]. This allows for the printing of 3D hydrogels with slower gelation rates while maintaining quality [144]. However, there is a limit that the viscosity of a bioink can reach as dictated by properties of the polymers. High viscosity increases the shear stress, resulting in cell damage (especially in 3D bioprinting) and causes extrusion difficulties [145,146,147]. Therefore, the viscosity of the bioink must be optimized. Also, factors such as pressure, feed rate, print head speed, and printing distance affect the printability of hydrogels [148].

When cells are printed along with the materials, those materials are termed bioinks to distinguish them from the printing process where cells are added post-printing. An ideal bioink for extrusion-based printing should have good printability, sufficient mechanical strength to support the printed structure, high interfacial strength between the printed layers to avoid delamination, and favorable conditions for cell loading. Most conventional hydrogels are weak and brittle, making them difficult to handle during printing. Moreover, these hydrogels cannot retain the shape integrity and fidelity of the print [133]. To mitigate those problems, two approaches are considered: (i) hydrogels with high strength and elasticity are made as interpenetrating networks, composites with nanomaterials, and single-network polymer chains with covalent crosslinking, and (ii) using a bath of sacrificial reversible hydrogels such as poloxamer into which hydrogels are printed.

Direct ink writing (DIW) is a material extrusion technique in which ink in a liquid phase is extruded through a nozzle in a predefined path to fabricate a 3D structure. This technique can work with a wide range of hydrogels and is capable of printing multiple hydrogels simultaneously [149]. However, depositing hydrogel precursors is difficult. Therefore, the viscosity of hydrogels is increased by pre-crosslinking [150,151,152] or adding nanoclays [149]. Pre-processing has a significant drawback as it affects the mechanical properties of the printed gel. The print speed of the nozzle movement and the extrusion should be controlled to work with the viscosity of the hydrogel precursor [152]. For example, a print speed that is too fast relative to the volume flow rate of the hydrogel could stretch the polymers in the hydrogel, affecting the mechanical properties of the printed structure. When chitosan–gelatin was printed at high print head speeds, there was no formation of continuous fibers due to excessive stretching of the microstructure [82]. To mitigate self-assembly of chitosan into hydrogels, some have explored alternative solvents [118].

Extrusion-based techniques usually have low resolution. In contrast, light-based methods are faster and have much higher resolution [153,154]. Digital projection lithography (DLP) and stereolithography are examples of light-based 3D printing. However, these techniques can only print photopolymerizable hydrogels [155].

An ideal bioink should be bioprintable, biocompatible, minimally cytotoxic, capable of encapsulating cells, and maintain its printed shape under wet conditions [154,156]. The hydrogels used for 3D bioprinting should meet the printing process requirements while keeping cells alive during and after the process. Since 3D bioprinting involves live cells, the process needs to be gentle. Therefore, not all 3D printing techniques will work. For example, methods that extrude materials at elevated temperatures are not suitable for bioprinting. FDM is such a technique that requires materials to be extruded at 140–250 °C [157]. The standard methods used in 3D bioprinting are DIW and inkjet bioprinting.

### 3.4. Current Developments of 3D Printing of Hydrogels

Recent studies of the 3D printing of hydrogels focused on developing the hydrogel material for targeted applications. For example, the antimicrobial properties of hydrogels were achieved with the addition of silver nanoparticles [158,159]. Moreover, the introduction of nanoparticles leads to an improvement in mechanical properties, i.e., the introduction of nanocrystalline cellulose to alginate [158] and nanodiamonds to hyaluronic acid hydrogels [160].

Some scaffolds prepared using combinations of natural materials and different processing techniques have been shown to provide a conducive environment. However, the same combinations may not meet the requirement for 3D printing where a solution has to transform into a solid. Current 3D printing techniques use a layer-by-layer approach to print an object. Layer-by-layer fabrication has drawbacks such as constrained geometric capabilities, degraded surface quality, increased postprocessing, limited throughput, and anisotropy in mechanical performance [161]. These weaknesses can be overcome by volumetric additive manufacturing technology. The two main methods used in volumetric 3D printing are two-photon photopolymerization and computed axial lithography. However, the volume generation rate of two-photon photopolymerization is low [162]. In computed axial lithography, light energy is delivered from different angles to a rotating resin as a set of images [161]. Another method for 3D volumetric printing was introduced recently, known as xolographic 3D printing [163]. The xolographic 3D printing technique uses dual-color photoswitchable photoinitiators added to a resin, where the first wavelength initiates the photopolymerization, and initiation or inhibition is realized by the second wavelength. This technique can lead to a resolution ten times higher than computed axial lithography. However, the use of this technology on hydrogels is yet to be subjected to experimentation.

Usually, light-based 3D printing techniques use UV/violet light. A new approach was recently introduced to replace UV/violet light with mild and tunable visible wavelengths [164]. High energy wavelengths may have adverse effects on hydrogel systems, i.e., cells, biological molecules, nanocomposites, and multi-material structures. Therefore, it is beneficial to polymerize the hydrogel under mild conditions. This new technique uses panchromatic photopolymer resins. Since this technique is novel, the feasibility of using this method to print hydrogels must be evaluated.

As mentioned in previous sections, current methods used for the 3D printing of hydrogel have weaknesses. Therefore, studies have been conducted to improve the existing methodologies. Table 2 summarizes some of the recent improvements in 3D printing techniques.

## 4. Advantages of 3D Printing over Conventional Fabrication Methods

The traditional methods used in synthesizing hydrogels include polymerization methods such as free-radical polymerization and physical or chemical crosslinking methods. Moreover, methods including but not limited to emulsification, gas foaming–leaching, electrospinning, photolithography, and micro molding are involved in the fabrication of hydrogels with specific morphology [23]. However, the conventionally used hydrogel fabrication methods are dependent on application. Section 2.2, Section 2.3 and Section 2.4 describe the conventional methods used to fabricate hydrogels based on the applications and their limitations when considering biomedical applications. In the biomedical field, 3D printing has many benefits. For example, enabling personalized medicine involving custom-made medical products and equipment, cost-efficiency (for small-scale productions) due to minimal use of resources, fast—printing takes a few hours—and easy to use, not requiring high expertise knowledge to handle [173]. 3D printing can fabricate anatomical models, which are helpful for surgical preparations and studies. Anatomical models allow doctors to gain insight into a patient’s specific anatomy to determine the best surgical procedures [174]. A recent study generated a heterotypic 3D model by combining stromal cells, their extracellular matrix, and parenchymal epithelial cells [175]. The extracellular matrix was constructed using extrusion-based 3D printing using peptide-modified alginate hydrogels as the material. This model can be used to study breast stroma–parenchyma interactions. Most conventional methods used in the scaffold formation, e.g., gas foaming, freeze-drying, and salt leaching, have poor biofunctionality, which may be due to poor cell homogeneity, lack of vascularization networks, and the presence of a necrotic core. However, 3D printing can produce scaffolds with more accurate architecture, high porosity, better pore size, and high reproducibility [176,177,178]. In 3D printing, the design can be separated from the manufacturing capabilities, giving researchers more freedom for collaboration and data sharing [179].

## 5. Challenges

In 3D printing, polymers/hydrogels are processed to construct the desired shape, especially in extrusion-based systems. Unfortunately, processing materials may affect the properties of the final product; increasing the processing speed may trigger rheological phenomena of polymers. Weak interlayer adhesion is another challenge in 3D printing as it influences the mechanical properties of the 3D print [85]. Dynamic hydrogels are used as a solution [180]. Dynamic hydrogels can be reformed by removing the shear force while preserving their structural integrity [181].

There are many 3D printers which offer different printing strategies, and each have different modes of controlling the fiber size and placement of cells. Some of the parameters that have been evaluated for a few combinations of materials are print-head speed, polymer concentration, and nozzle size. Also, there are different mechanisms through which hydrogel printing is obtained which further affects the formed fibers. Many studies list applied pressure as an indicator during the extrusion-based approach. However, applied pressure is a function of the polymer viscosity, nozzle size, and type of extruder used. Rather, it would be advantageous to provide a volume flow rate or a relationship between applied pressure and printing material flow rate. Volume flow rate information could then be used to determine shear stresses and understand the effect of 3D printing on cellular components. Also, one could develop models that can predict fiber characteristics. Much of the model development and hydrogel formulations are focused on improving the mechanical properties. However, the effect of microstructure of the hydrogel on cells during bioprinting needs further attention. These require better understanding of the changes in the microstructure of the hydrogel due to various printing parameters. Using soy protein isolates as printing materials along with husk and apple fibers, the influence of microstructure on printing conditions was evaluated [182]. These results showed the effect of the distribution of fibers and protein on the porosity of the formed fibers. Many properties related to the transport of molecules in the printed fibers have not been explored, particularly when cells are printed along with the fibers. Optimizations of print size, mechanical properties, and combinations are emerging. To enhance the properties of printed structures, aligning the fibers made of synthetic polymers has been explored using print parameters [183]. Some have explored hierarchical alignment based on the microstructure of the 3D-printed structures [184,185]. Although such attempts are helpful in 3D bioprinting, the diffusion of nutrients to cells colonizing inside the fibers has not been explored.

Another issue of extrusion-based printing is the resolution (100–400 μm) [186]. Though some strategies have been implemented, the expected resolution has still not been achieved. In contrast, stereolithography can produce prints with high resolution (10–150 μm) [186]. Even though soft lithography is not a 3D printing technique, soft lithography can build 3D structures with much higher resolution (<0.01 μm) [186]. Despite the low resolution, additive manufacturing has more advantages than soft lithography. However, maintaining the high resolution and structural accuracy of a print is crucial. Therefore, achieving higher resolution than soft lithography for 3D printing technology is essential. Other drawbacks of stereolithography are high printing costs, slow printing times, and the inability to structure multiple materials simultaneously. The development of a functional bioink is a challenge of 3D bioprinting. A successful bioink should facilitate the viability of cells, maintaining optimal conditions, such as temperature, pH, osmolarity, physical forces, and pressure. It is essential to maintain desired mechanical properties and provide a dynamic structural environment at the cellular level to mimic the actual environment of the extracellular matrix of a tissue. However, such an optimal hydrogel bioink has not yet been formulated [187]. One of the biggest challenges in developing a fully functional bioink is to match the properties of the ink with the requirements of the 3D printing techniques. For example, extrusion-based printers require bioinks with shear-thinning properties, and inkjet printers require rapid crosslinking [188,189]. Maintaining the consistency of the 3D bio print is highly challenging, mainly when natural polymers are used. Natural polymers have varying molecular weights and compositions, producing low batch-to-batch consistency [187].

## Figures and Tables

**Figure 1 gels-11-00192-f001:**
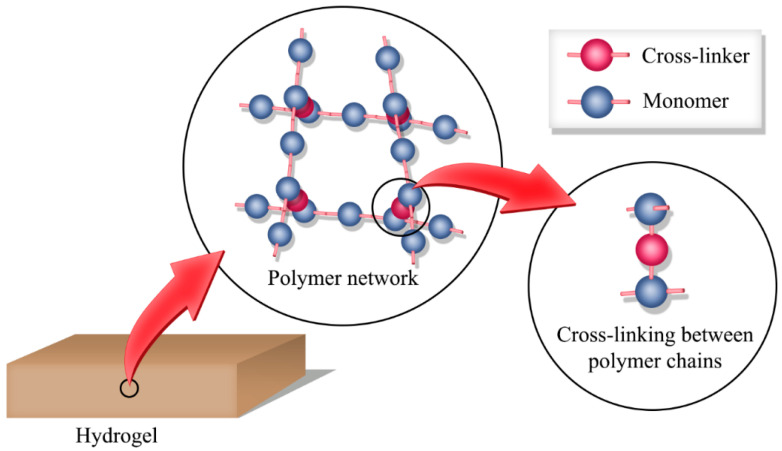
The structure of a hydrogel: polymer network with crosslinks.

**Figure 2 gels-11-00192-f002:**
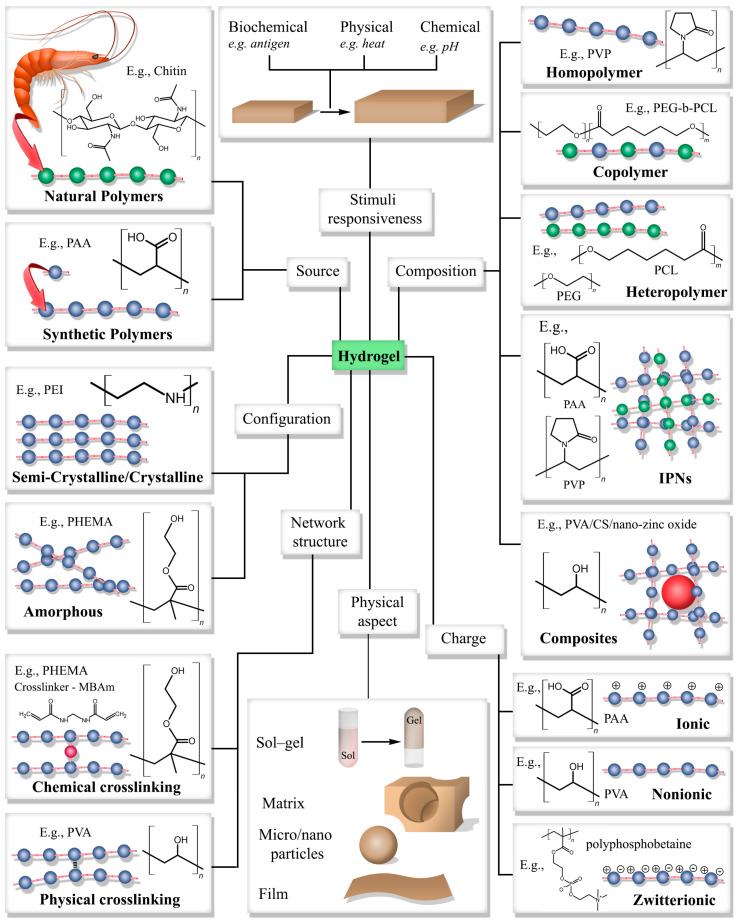
A classification scheme for hydrogels. Here, the beads and strings represent the polymers used to produce hydrogels. The blue and green spheres represent different monomer units, and the bond between monomers is shown in pink. The red sphere represents a crosslinker. The dashed line represents the intermolecular interactions that lead to the formation of physical crosslinks. The large red sphere represents a composite material such as a nanoparticle.

**Figure 3 gels-11-00192-f003:**
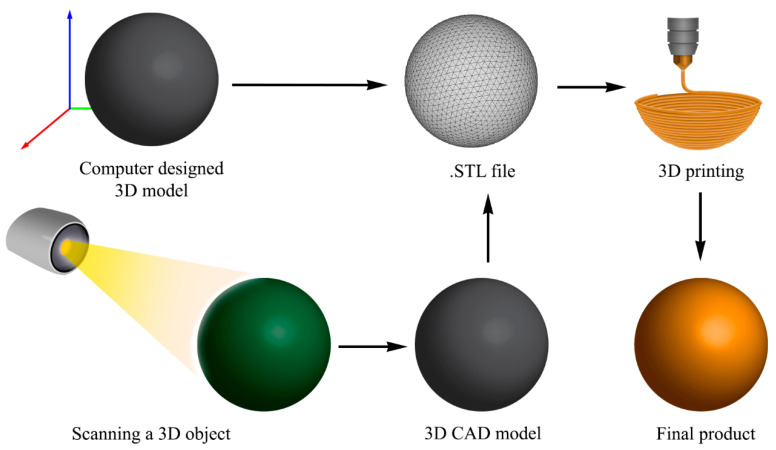
General steps of 3D printing. The 3D object can be scanned using a 3D scanner or designed on a computer.

**Figure 4 gels-11-00192-f004:**
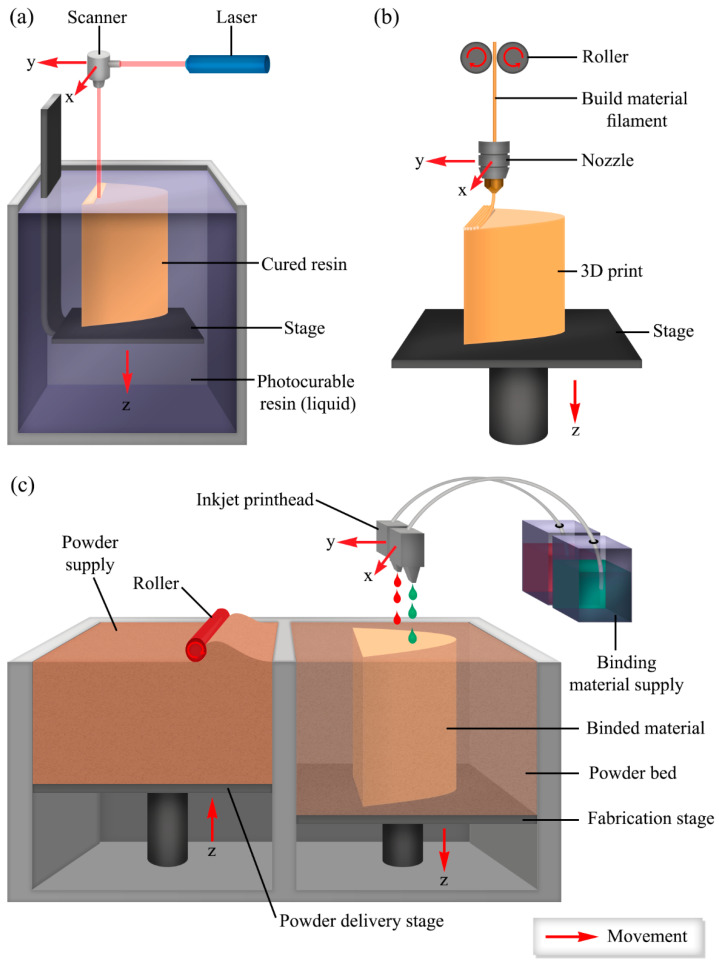
Common 3D printing techniques used for printing hydrogels; (**a**) laser-based systems (e.g., stereolithography); (**b**) extrusion-based techniques (e.g., FDM); and (**c**) inkjet printing (e.g., binder jetting).

**Table 2 gels-11-00192-t002:** Summary of recent improvements in 3D printing techniques.

Hydrogel	3D Printing Technique	Improvements in the 3D Printing Process	Targeted Application	Ref.
PEGDA	Stereolithography	High print resolution with water-soluble photo blockers that absorb violet light (chlorophyllin and tartrazine)	Applications involve adding living cells	[165]
PNIPAM, PEGDA, PAMPS, and PAAm	Capacitor edge effect	Liquid precursors are patterned and then polymerized—high resolution and applicable for a wide range of hydrogels	Artificial tissues, soft metamaterials, soft electronics, and soft robotics	[166]
OHA/GC/ADH	Extrusion-based 3D bioprinting	Self-healing properties—correct the gel fracture due to high shear stresses applied in the extrusion-based printing. Also, polymer concentration and molecular weight of HA is controlled to tailor viscoelastic properties of the hydrogel	Tissue engineering—cartilage regeneration	[167]
HAMA, GelMA, and alginate	Direct extrusion printing, sacrificial printing, and microfluidic hollow fiber printing	Post-treatment of the printed structures by immersing in a polycationic chitosan solution—complexation-induced resolution enhancement		[168]
HA-g-pHEA-Gelatin	Extrusion-based 3D bioprinting	Improved hydrogel’s mechanical stability	Tissue engineering	[169]
Silk fibroin hydrogel	DLP—for 4D printing	Shape morphing of a bilayer hydrogel (by anisotropic volume change) to overcome the limitation of DLP printing to fabricate obvolute structures with two or more components	Tissue mimetic scaffolds	[170]
Alginate-based hydrogels	Micro-extrusion process	Incorporation of graphene oxide into the hydrogel inks—improved shape fidelity and resolution	Tissue engineering	[171]
Agar/calcium alginate	Extrusion based printing	Introduction of agar—improved resolution and higher precision	Artificial tissues	[172]

## Data Availability

No new data were created or analyzed in this study. Data sharing is not applicable to this article.

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
