# Peer review of "3D-Printed Hydrogels from Natural Polymers for Biomedical Applications: Conventional Fabrication Methods, Current Developments, Advantages, and Challenges"

_gels, 2025, doi:10.3390/gels11030192_

Round 1

Reviewer 1 Report

Comments and Suggestions for Authors

It is a review paper. The authors addressed the review of 3D printing of hydrogels. The topic is relevant to 3D painting, which is now emerging as a versatile manufacturing technique. This review paper provides an overview of hydrogels printing. In my opinion, the topic is worth investigating. It will not add anything new, but it is extremely useful for young researchers to understand developments in this field. The references are appropriate, tables and figures are ok.

The current review focuses on 3D printing of hydrogels. The paper contains useful information. Following suggestions can be considered

1.       Please elaborate more on rheological properties, and their significance while printing by DIW.

2.       The application part in this manuscript is weak. Further explanations can be given about applications with illustrations.

3.       Are there any modeling results available that focuses on the process simulation? Please include them.

Comments on the Quality of English Language

Few spelling mistakes are detected. Please spell check the entire manuscript one more time. 

Author Response

It is a review paper.  The authors addressed the review of 3D printing of hydrogels. The topic is relevant to 3D painting, which is now emerging as a versatile manufacturing technique. This review paper provides an overview of hydrogels printing.  In my opinion, the topic is worth investigating. It will not add anything new, but it is extremely useful for young researchers to understand developments in this field. The references are appropriate, tables and figures are ok.

The current review focuses on 3D printing of hydrogels. The paper contains useful information. Following suggestions can be considered

  1. Please elaborate more on rheological properties, and their significance while printing by DIW.

Response: Thank you.  As suggested, we have elaborated on the rheological properties.

  1. The application part in this manuscript is weak. Further explanations can be given about applications with illustrations.

Response:  There are many review articles which focus on applications.  We have provided some links to those rather than repeat the content

  1. Are there any modeling results available that focuses on the process simulation? Please include them.

Response: This review is written with the focus of researchers who want to get to this area as a foundational paper. T here are modeling strategies for predicting the fiber sizes.  However, we believe that is not the focus of this article.    

Few spelling mistakes are detected. Please spell check the entire manuscript one more time.

Response:  Thank you.  We have gone through the entire manuscript to eliminate typographic errors.

Reviewer 2 Report

Comments and Suggestions for Authors

In this manuscript, the authors present a comprehensive review of 3D-printed hydrogels derived from natural polymers for biomedical applications. The manuscript offers an overview of the fundamental of hydrogels and 3D printing technology, explores recent advancements in 3D printing for fabricating natural hydrogels, discusses the advantages of utilizing 3D printing over conventional methods for hydrogel production, and the challenges associated with 3D printing. Overall, this manuscript is well-prepared. I recommend acceptance after addressing the following issues:

1.     The "Gelation Mechanism" column in Table 1 should be reorganized for clarity. Entries such as “Blending with HA,” “GO as additive,” “HAp as additive,” and “Dissolved in Alkali/Urea aqueous solution” do not represent gelation mechanisms. As well, “Thermal gelation,” “UV crosslinking,” and “Photocrosslinking” are methods that trigger crosslinking.

2.     In Table 1, the "Gelatin Concentration" column contains entries such as “1g” and “6 mg,” are not concentration.

3.     In Table 2, the authors should present a more representative and structured summary of how different material designs enhance the 3D printing process. For example, the description “improved hydrogel’s mechanical stability” for HA-g-pHEA-Gelatin hydrogels is too vague and lacks sufficient detail. Similarly, in “Alginate-Based Hydrogels” and “Agar/calcium alginate,” the addition of grapheme oxide and agar is mentioned as enhancing resolution, but these examples have not been effectively categorized or differentiated. A clearer and more structured summary should be based either on material design strategies or the specific advancements aligned with different 3D printing techniques.

4.     In Section 4, “Advantages of 3D Printing over Conventional Fabrication Methods,” the authors could provide a brief overview of conventional hydrogel fabrication methods commonly used in the biomedical field. This addition would help readers quickly grasp the current state and challenges associated with these traditional methods, thereby offering a clearer context for understanding the benefits and advancements offered by 3D printing in the following examples..

5.     Some formatting issues, including inconsistencies in format and capitalization.

Line 237, Figure 3” is bold, whereas other instances of “Figure” are not.

Line 343, the first letter of “Alginate” is incorrectly capitalized.

Author Response

In this manuscript, the authors present a comprehensive review of 3D-printed hydrogels derived from natural polymers for biomedical applications. The manuscript offers an overview of the fundamental of hydrogels and 3D printing technology, explores recent advancements in 3D printing for fabricating natural hydrogels, discusses the advantages of utilizing 3D printing over conventional methods for hydrogel production, and the challenges associated with 3D printing. Overall, this manuscript is well-prepared. I recommend acceptance after addressing the following issues:

  1. The "Gelation Mechanism" column in Table 1 should be reorganized for clarity. Entries such as “Blending with HA,” “GO as additive,” “HAp as additive,” and “Dissolved in Alkali/Urea aqueous solution” do not represent gelation mechanisms. As well, “Thermal gelation,” “UV crosslinking,” and “Photocrosslinking” are methods that trigger crosslinking.

Response:   Thank you.  We have edited them to reflect the gelation mechanism as well as made them consistent with other tables.

  1. In Table 1, the "Gelatin Concentration" column contains entries such as “1g” and “6 mg,” are not concentration.

Response:  As suggested, we have made changes to inconsistencies in writing the concentrations

  1. In Table 2, the authors should present a more representative and structured summary of how different material designs enhance the 3D printing process. For example, the description “improved hydrogel’s mechanical stability” for HA-g-pHEA-Gelatin hydrogels is too vague and lacks sufficient detail. Similarly, in “Alginate-Based Hydrogels” and “Agar/calcium alginate,” the addition of grapheme oxide and agar is mentioned as enhancing resolution, but these examples have not been effectively categorized or differentiated. A clearer and more structured summary should be based either on material design strategies or the specific advancements aligned with different 3D printing techniques.

Response:   Thank you for the suggestion.  The problem is there are too many parameters to categorize and rationalize.  In order to promote more consistency, one needs some common factor such as volume flow rate.  We have added few lines in the challenges in this regard.

  1. In Section 4, “Advantages of 3D Printing over Conventional Fabrication Methods,” the authors could provide a brief overview of conventional hydrogel fabrication methods commonly used in the biomedical field. This addition would help readers quickly grasp the current state and challenges associated with these traditional methods, thereby offering a clearer context for understanding the benefits and advancements offered by 3D printing in the following examples.

Response:   Thank you.  We described conventional fabrication methods in earlier sections.  We have added a few more lines describing those in the section.

5.Some formatting issues, including inconsistencies in format and capitalization.

Line 237, “Figure 3” is bold, whereas other instances of “Figure” are not.

Line 343, the first letter of “Alginate” is incorrectly capitalized.

Response:  We appreciate thorough review. We have correct these errors.

Reviewer 3 Report

Comments and Suggestions for Authors

In this manuscript, authors introduce classification of hydrogels, common fabrication techniques for producing hydrogel structures, 3D printing techniques for structure control of hydrogels. However, it is not very convincing when authors try to illustrate structural complexities which limit the use of hydrogels and why 3D printing can definitely overcome the problem in addition to geometry control and resolution control. It is recommended the manuscript be reviewed after major revision.

More references is needed for the Introduction and 2.Hydrogels section.

Figure 1 is not clear. Please refine the pattern in the middle circle to conform with the zoom-in circle.

For Figure 2, it may be better to give chemical structure examples for categories including source, configuration, charge. It’s hard to get a sense of what the chemical structures look like by just seeing positive/negative charge symbols or connection of the shapes/patterns.

Please add Figures for each subsection under 2.Hydrogels.

For 2.2.2. Fabrication Methods of Hydrogels section, authors first discussed how to create pores in hydrogels, then how to make patterns in hydrogels, followed by hydrogels with specific designs and nanogels/microgels; this is hard to link with the following 3D printing section because it’s very confusing about the center point of the comparison. Do the authors want to compare geometry control through 3D printing to traditional hydrogel macroscopic structure-control techniques?

Authors also spent long paragraphs on Hydrogels in Biomedical Applications and Limitations of Hydrogels; however, it is not convincing that 3D printing can meet majority of the needs or resolve majority of the issues. There are more things about microscopic chemical structures and the resulting macroscopic physical properties.

When demonstrating different 3D printing techniques, please give either schematic illustration or examples. e.g., it would be easier for readers to understand if they can see how the extrusion-based DIW works.

Author Response

In this manuscript, authors introduce classification of hydrogels, common fabrication techniques for producing hydrogel structures, 3D printing techniques for structure control of hydrogels. However, it is not very convincing when authors try to illustrate structural complexities which limit the use of hydrogels and why 3D printing can definitely overcome the problem in addition to geometry control and resolution control. It is recommended the manuscript be reviewed after major revision.

More references is needed for the Introduction and 2. Hydrogels section.

Response:  As suggested we have added nearly 25 extra references to the manuscript.

Figure 1 is not clear. Please refine the pattern in the middle circle to conform with the zoom-in circle.

Response:  As suggested, we have edited Figure 1.

For Figure 2, it may be better to give chemical structure examples for categories including source, configuration, charge. It’s hard to get a sense of what the chemical structures look like by just seeing positive/negative charge symbols or connection of the shapes/patterns.

Response:  Thank you.  There are many chemicals in each category.  Our intent was to generalize the categories while providing description to those. 

Please add Figures for each subsection under 2.Hydrogels.

Response:  We are not sure what other figures the reviewer wanted us to add

For 2.2.2. Fabrication Methods of Hydrogels section, authors first discussed how to create pores in hydrogels, then how to make patterns in hydrogels, followed by hydrogels with specific designs and nanogels/microgels; this is hard to link with the following 3D printing section because it’s very confusing about the center point of the comparison. Do the authors want to compare geometry control through 3D printing to traditional hydrogel macroscopic structure-control techniques?

Response:  Thank you.  There are many chemicals in each category.  Our intent was to generalize the categories while providing description to those. 

Authors also spent long paragraphs on Hydrogels in Biomedical Applications and Limitations of Hydrogels; however, it is not convincing that 3D printing can meet majority of the needs or resolve majority of the issues. There are more things about microscopic chemical structures and the resulting macroscopic physical properties.

Response:  Thank you.  We agree with the reviewer there are more aspects to chemical structures.  We have written exclusively on various chemical modifications in hydrogels.  (Bandarage and Madihally, J Applied Polymer Science 2021) We did not want to repeat those in this review while providing a link to that review. 

When demonstrating different 3D printing techniques, please give either schematic illustration or examples. e.g., it would be easier for readers to understand if they can see how the extrusion-based DIW works.

Response:  Thank you.  We believe there are several reviews describing different 3D printing techniques.  We purposefully avoided those description while focusing on hydrogels in 3D printing.

Round 2

Reviewer 2 Report

Comments and Suggestions for Authors

The authors have successfully addressed all major concerns, I support its acceptance for publication.

Author Response

The authors have successfully addressed all major concerns, I support its acceptance for publication.

Response: Thank you.  We appreciate the comments from reviewer.

Reviewer 3 Report

Comments and Suggestions for Authors

Generalizing the categories without providing intuitive chemical structures is not easy for the readers to understand.

It is not convincing that 3D printing can meet majority of the needs or resolve majority of the issues authors mentioned. There are more things about microscopic chemical structures and the resulting macroscopic physical properties. If authors believe this has been illustrated in previous publications. They should mention this and give a summary in the manuscript and show why this manuscript is significant compared to others.

Author Response

Generalizing the categories without providing intuitive chemical structures is not easy for the readers to understand.

Response:   As suggested, we have updated Figure 2 providing example chemicals and their structures.

It is not convincing that 3D printing can meet majority of the needs or resolve majority of the issues authors mentioned. There are more things about microscopic chemical structures and the resulting macroscopic physical properties. If authors believe this has been illustrated in previous publications. They should mention this and give a summary in the manuscript and show why this manuscript is significant compared to others.

Response:   Thank you.  We revised Section 3.3 addressing the microstructure and its role in 3D printing.  We also edited Section 5 adding comments on possible limitations of 3D printing.  These required seven new references added to the review.